# Impact of Friendship Bench problem-solving therapy on adherence to ART in young people living with HIV in Zimbabwe: A qualitative study

**Ilhame Ouansafi**[1]*, **Dixon Chibanda**[2,3,4], **Epiphania Munetsi**[2,5], **Victoria Simms**[6]

**1** London School of Hygiene and Tropical Medicine, London, United Kingdom, **2** Friendship Bench, Harare, Zimbabwe, **3** Department of Psychiatry, College of Health Sciences, University of Zimbabwe, Harare, Zimbabwe, **4** Centre for Global Mental Health, London School of Hygiene and Tropical Medicine, London, United Kingdom, **5** Zimbabwe AIDS Prevention Project, College of Health Sciences, University of Zimbabwe, Harare, Zimbabwe, **6** Medical Research Council Tropical Epidemiology Group, London School of Hygiene and Tropical Medicine, London, United Kingdom

* ilhame.ouansafi1@alumni.lshtm.ac.uk

## Abstract

### Background

Adolescents and young people globally are highly vulnerable to poor mental health especially depression, and they account for 36% of new HIV infections in Eastern and Southern Africa. HIV services remain inadequate for this population and their adherence to ART is low. The Friendship Bench (FB), an evidence-based model developed in Zimbabwe to bridge the mental health gap, is a brief psychological intervention delivered on benches in primary care facilities by lay health workers ("grandmothers") trained in problem-solving therapy. This study explored the experience of young people living with HIV attending FB, and their perception of how problem-solving therapy impacted their adherence to ART.

### Methods

Semi-structured interviews were conducted in July 2019 with 10 young people living with HIV aged 18–24 years, who had recently completed FB counselling in Harare. Participants were purposively sampled and recruited from three primary care facilities. Interviews were conducted in Shona, audio-recorded, transcribed verbatim and translated into English. Transcripts were analysed in NVivo12 using inductive thematic analysis.

### Results

Study findings revealed a clear emotional denial towards HIV, particularly for young people infected perinatally, and a resulting low adherence to ART. The study also unpacked the issues of internal stigma and how young people living with perinatally acquired HIV are informed of their HIV status. Participants reported that FB had a critical role in helping them accept their HIV status. Grandmothers' empathic attitude was key during counselling on adherence to ART, to demystify the disease and treatment, normalize the reality of living

**Data Availability Statement:** The interview transcripts contain sensitive patient information related to their mental wellbeing including harm and suicidal thoughts, personal and family issues,

including physical and emotional violence. It is not possible to de-identify the participants due to the small sample size and detailed history collected. The Medical Research Council of Zimbabwe (ref MRCZ/A/2130) and the London School of Hygiene and Tropical Medicine Research Ethics Committee (ref 16708) granted ethical approval for the study based on data protection. The data underlying the results presented (i.e. interviews' transcripts) are available at https://datacompass.lshtm.ac.uk/ and data requests may be sent there.

**Funding:** This study was supported by a 2018-2019 David Bradley Travel Grant and 2018-19 LSHTM School Trust Funds awarded to IO to pay for part of her travel expenses. No additional external funding was received for this study.

**Competing interests:** The authors have declared that no competing interests exist.

with HIV, encourage young people to socialize with peers and free them of guilt. Interviewees unanimously reported improved ART adherence following FB counselling, and many described enhanced health and wellbeing.

## Conclusion

Participants saw FB as a strong contributor to their general well-being, evident in decreased symptoms of depression and improved adherence to ART. FB problem-solving therapy should be rolled out to further support young people after post-test counselling or HIV serostatus disclosure for perinatally acquired HIV, for acceptance of HIV status and adherence to ART.

## Background

Eastern and Southern Africa accounts for 54% of the estimated 37.9 million people of all ages living with HIV globally in 2018. About 36% of the estimated 800,000 new HIV infections in this region occurred among adolescents and young people aged 15–24 years. Of the 3.5 million young people living with HIV (YLHIV) globally, 80% are in sub-Saharan Africa and 63% in Eastern and Southern Africa alone [1, 2]. Zimbabwe has one of the highest adult HIV prevalence in the region (12.7%), in 2018 there were about 1.3 million people living with HIV (PLHIV), including 84,000 children aged 0–14 years and 130,000 young people aged 15–24 years [2, 3]. Despite the high vulnerability to HIV infection among adolescents and young people, rates of HIV diagnosis and treatment initiation remain very low, and their low adherence to ART is particularly concerning [4]. In Zimbabwe, the prevalence of viral load suppression is 48.6%/ 40.2% among young women/men aged 15–24 years living with HIV, compared to 78.7%/71.1% respectively for adults aged 45–54 years [5].

About 75% of the people affected by mental disorders are in low-income countries where limited access to appropriate treatments [6, 7], lack of resources, lack of trained health-care providers and social stigma associated with mental disorders are all barriers to effective mental care [8]. Depression is a common mental disorder affecting over 264 million people globally [9]; it is the leading cause of disability worldwide and a significant contributor to the global burden of disease [10]. Adolescents and young people are particularly vulnerable to poor mental health [11]. Several health, development and cultural factors are associated with mental disorders in this population, including educational achievements, substance use and abuse, violence, child abuse, reproductive and sexual health, poverty and social disadvantage [11]. An estimated 10–20% of adolescents globally experience mental health disorders, depression being one of the leading causes of illness and disability in this age group, and suicide the second leading cause of death [12].

Depressive symptoms are common among PLHIV [11, 13]. Recent studies in Zimbabwe and Tanzania explored the mental health and lived experiences of YLHIV. Psychosocial challenges identified include loss and grief, chronic domestic abuse, financial stressors, internalized and community stigma, difficulties in accepting HIV status, self-blame, low self-worth, isolation and rejection [14–16]. Studies have shown that depression is associated with nonadherence to ART [17] and that treatment of depression through antidepressants and/or psychotherapy improves adherence to ART [18] Wider public health approaches to prevent depression in PLHIV are needed, because of the negative impact on adherence to ART, disease progression and mortality [19]. A study among adolescents living with HIV in Zambia

highlighted challenges to ART adherence, including loss of a mother, lack of knowledge about HIV, and psychosocial distress [20].

The Friendship Bench (FB) is an innovative model developed in Zimbabwe to bridge the gap in mental health treatment. Embedded within the City Health Department of Harare, it offers problem-solving therapy delivered on benches in primary care facilities by trained lay health workers (LHWs), elderly women commonly known as community "grandmothers". After a cluster randomized controlled trial confirmed the effectiveness of the model at improving mental health [21, 22], it was scaled up in primary health care clinics across Harare and has been replicated several countries. In 2019, FB was formally endorsed by the Ministry of Health as a national programme [23]. FB counselling consists of six sessions generally completed within four to six weeks. LHWs ask questions, encourage clients to "open their minds", identify a problem and proactively tackle it. Following problem identification and exploration, LHWs guide their clients on an action plan towards a feasible solution [24, 25]. FB is widely accepted in the communities, benches are quite public and viewed as a therapeutic environment [26]. Before starting the FB counselling, clients are assessed by LHWs using the Shona Symptoms Questionnaire (SSQ), a locally developed and validated 14-item measure of common mental disorders. The SSQ is widely used in Zimbabwe as a screening tool with reliable sensitivity and specify [27]. A score equal or above 9 on the SSQ indicates a risk of common mental disorder [21].

## Study rationale

This qualitative study aimed to understand some of the psychosocial factors contributing to nonadherence to ART among YLHIV aged 18–24, and to explore their experiences and perceptions accessing FB on how problem-solving therapy impacted their adherence to ART. FB was not designed specifically for PLHIV, and it is therefore important to understand clients' experiences linked to their particular context living with HIV and their perceptions on any impact FB has on their adherence to ART. This will help assess the quality of services, effectiveness and acceptability given the high prevalence of HIV in Zimbabwe, and identify opportunities and areas of improvement as FB continues to expand.

## Methods

### Setting

Data collection took place in Harare in July 2019. Eligible participants were recruited from three primary care clinics offering FB counselling services and receiving a large proportion of adolescent patients. Interviews were conducted away from the clinic in the administrative office of FB to ensure participants' privacy.

### Study design and sampling

To address the research question, semi-structured interviews were conducted with young people, aged 18–24 years, living with HIV and having been prescribed ART, who had completed the FB counselling in the last year, after initially scoring 9 or above in the SSQ-14. Purposive sampling was used to select 22 people based on the characteristics aforementioned, who were contacted by the LHWs. Seven were not available during the time of interviews and 5 did not meet the study inclusion criteria, and were thus excluded. The remaining 10 people were interviewed.

## Data collection

Interviews were carried out by two Zimbabwean research assistants at FB, who were trained in qualitative interviewing. The topic guide (S1 and S2 Files) was developed in English by the lead author, discussed with Zimbabwean co-authors to ensure cultural compliance, then translated into Shona by the research assistants. Interviews explored (i) the participants' experience about finding out HIV status, (ii) what triggered them to visit FB and their initial expectations, (iii) their overall experience through the FB counselling and their perception of any changes in their mental well-being and quality of life, (iv) their knowledge of ART and adherence, (v) difficulties they were experiencing with adherence to ART and their perception of any changes during/following the FB counselling.

## Data analysis

Interviews were audio-recorded, transcribed verbatim in the original language (mostly Shona with some words/expressions in English) then translated to English. The lead author reviewed the transcript and conducted an inductive thematic analysis, with the aim to provide a comprehensive summary of the psychosocial factors contributing to nonadherence to ART, the overall participants' experience of the FB counselling, and the perception of any impact on their adherence to ART. Initial coding was done on hard copies of the transcripts, recurrent themes and sub-themes were identified, and a preliminary coding framework was developed. A more detailed line-by-line coding was completed using NVivo 12. The initial coding framework was adapted: themes were created, merged or deleted as appropriate, to best reflect ideas extracted from the transcripts, and grouped into key thematic areas. This iterative process added to the credibility of the study. Results were discussed and confirmed with the research assistants and co-authors.

## Ethical considerations

The Medical Research Council of Zimbabwe (ref MRCZ/A/2130) and the London School of Hygiene and Tropical Medicine Research Ethics Committee (ref 16708) gave ethical approval for the study.

Written informed consent was obtained from all participants and the signed consent forms are stored in a locked cabinet at the FB administrative office. Transcripts of the interviews have been anonymised and soft copies stored in an encrypted USB. Participants were reimbursed for their participation in accordance with local regulatory research and ethics bodies.

# Results

The sample interviewed (Table 1) was 70% female, and 90% of all participants acquired HIV perinatally. The interviews lasted between 25 and 50 minutes.

Several themes emerged from the inductive thematic analysis of the transcripts and were categorised under four broader thematic areas (Table 2).

## Depression and living with HIV

MTCT was a recurring theme throughout the interviews. Most participants found out about their HIV status during adolescence, generally following a sickness and visit to health facilities, where they were offered HIV testing. Several expressed feelings of abandonment and lack of protection, especially those orphans raised by family members.

**Table 1. Descriptive characteristics of the participants.**

| Participants | Age (years) | Sex | SSQ week 1 | SSQ week 4 | Mode of HIV transmission | Years on ART |
|---|---|---|---|---|---|---|
| 01 | 24 | M | 13 | 6 | Vertical | 18 |
| 02 | 21 | F | 12 | 7 | Vertical | 11 |
| 03 | 20 | F | 9 | 4 | Horizontal (age 16) | 3 |
| 04 | 24 | F | 9 | 5 | Vertical | 3 |
| 05 | 24 | F | 11 | 5 | Vertical | 9 |
| 06 | 18 | M | 10 | 4 | Vertical | 18 |
| 07 | 18 | M | 12 | 4 | Vertical | 11 |
| 08 | 21 | F | 12 | 4 | Vertical | 16 |
| 09 | 20 | F | 10 | 5 | Vertical | 2 |
| 10 | 21 | F | 11 | 3 | Vertical | 5 |

*P02: how did mother go on about this, why didn't she go to get me protected [from HIV], what actually happened?*

*P04: What hurt the most was my mother wasn't there anymore, she left me this disease, and I could not get anyone to care for me*

Some interviewees had been taking ART for years unaware they were HIV positive, believing it was medication for other conditions, such as rash or sores. Others only started ART later and were generally sick throughout their childhood while ART naive.

*P04: Before I knew [that I was HIV-positive], I was someone who wouldn't go for more than two weeks without getting sick and admitted*

Participants expressed pain of finding out and living with HIV, using words such as hurt, despair, hopelessness. "Kufungisisa", literally translated as "thinking too much", is the

**Table 2. Thematic areas identified from the transcripts.**

| | |
|---|---|
| Depression and living with HIV | • Mother-to-child transmission of HIV (MTCT)<br>• HIV positive, but not knowing it<br>• It's too hard<br>• Isolation–Is HIV visible?<br>• Disclosure<br>• Stigma and discrimination |
| Nonadherence to ART | • Why should I take ART? I haven't done anything<br>• It's too painful<br>• I'd rather die–I will die anyway<br>• Depression and nonadherence<br>• Lack of information |
| FB counselling and acceptance of HIV status | • Grandmothers' empathy<br>• HIV is not visible<br>• It's not your fault<br>• You're not alone<br>• Like other people/disease<br>• It's not the end<br>• Improved knowledge of HIV and ART |
| FB counselling on adherence to ART | • Importance of punctuality and regularity<br>• Adherence for improved health<br>• Avoid "thinking too much" to not affect medication<br>• Acceptance of HIV status<br>• Improved adherence to ART and impact on health |

expression commonly used for depression in Shona. Two participants revealed thoughts of committing suicide upon finding out they acquired HIV perinatally.

> P03: I used to think too much (kufungisisa), I was in pain. In fact, I would always cry because it [being HIV-positive] was something that was hard to accept and even today it still hurts

> P07: l thought of committing suicide because it's hard to be told you are HIV [positive] and children of other people will be smart [clean] while you are not

Some participants mentioned lack of social interaction, deliberately isolating themselves from others, by fear of discrimination. One common concern was: can people see I am HIV positive?

> P10: I was afraid that others would say we don't play with someone who has HIV, I didn't know that they had no idea about it since I do not show any signs about being HIV positive

Disclosure of HIV status was an important issue for most participants and the distinction was made between people they could confide in and count on for support during difficult times, and people who would gossip and discriminate against them. One participant, a peer supporter with an NGO, felt it was important to disclose his status freely, so others can identify with him and ask for support.

> P04: I told them so that sometimes if I get sick, they will know where to start from for this kind of sickness, how to treat it

> P03: At times you might tell a person who will go around telling everyone else [gossiping] even if they are relatives. You know those you can tell and those you can't

> P01: Yes [I disclosed to] family, friends and even adolescents l work with because it's not a secret for me. Because if l hide it, no one will believe me. So if l tell them, they will see we are all the same

Fear of stigma and discrimination, and generally fear of gossiping, were recurrent throughout the interviews, when discussing disclosure, interactions with others or visits to the clinic.

> P06: I'm embarrassed to walk around with my pills from the clinic. I could meet important people and the pills go "kuchu kuchu" in the bottle, so at times I feel discouraged.

YLHIV were judgmental towards other people living with HIV, stating promiscuous behaviours.

> P08: I knew HIV was only for those who slept with many men, and I had never slept with anybody

## Nonadherence to ART

Most participants opened up about poor adherence to ART prior to joining FB. Some could not accept their HIV status at first, the stigma and pain associated with the need for a lifelong treatment were factors of delayed start or low adherence to ART.

> P05: I just thought that I had never been with a man before, so I told myself that God will Intervene so I will just leave it like that [stop the ART]

In some extreme cases, death seemed an alternative to facing reality.

*P08: I would skip telling myself I'd rather die*

Several interviewees associated low adherence to ART with depression, some of them opened up:

*P06: It disrupts a lot because you can forget to go to the clinic to get your medication, or forget the date that you must go to collect [the pills]*

*P10: I don't know, but when it was time for me to drink [the pills], l could just feel that l did not want*

Several participants said they joined FB to receive information about HIV and ART. One elaborated on this, based on his own lived experience and also talking on behalf of other YLHIV he interacts with in youth counselling groups as a peer supporter.

*P01: When you take your medication for the first time, you will be given counselling so it will still be fresh in your head that you have to take your medication. But as time goes on, you will be asking yourself why you are still taking the medication, so it will make you stop or skip taking your medication because you won't have full information on why you are taking it.*

## Friendship Bench counselling and acceptance of HIV

One common reason stated for starting FB counselling was the pain of accepting the HIV status, many saying they had not done anything to deserve to be HIV positive.

*P05: When I heard it, I didn't accept it, I just could not because I had never done anything that would expose me [to HIV] so it was hard*

Participants reported that FB had a critical role in helping them accept their HIV status, and evoked various components of the counselling which contributed to their acceptance of HIV and decreased symptoms of depression.

LHWS' kindness and empathy was mentioned throughout the interviews; participants highlighted the comforting feeling to have someone to confide in when in despair, a trusted and open person, who accepts them as they are.

*P09: The grandmothers make our brains to be calm and they will be taking you as their children, really advising in a way that what you will be thinking*

*P02: l felt relieved that l have people who can talk to me like this, as a person living with HIV, so it made me feel very free*

The FB counselling helped comfort and reassure those who were worried that HIV was noticeable and who were isolating themselves by fear of discrimination.

*P09: They helped me see that no one can see that l am positive. l had asked them if it could be seen that l am positive and they just told me that it was only me who could see it.*

It greatly helped to alleviate the feeling of guilt and self-blame associated with contracting HIV.

*P01: Stigmatisation made it hard because l used to think that it was my fault that am positive*

*P08: I would ask why am I taking pills, but I was then helped in knowing that I was born like that [HIV-positive], you got during breastfeeding*

Some interviewees suffered from being the only youth among adults in ART clinics, and were grateful to FB as they met or heard of other YLHIV going through similar struggles.

*P05: I have accepted it, because I've seen that I'm not the only one. There's quite a number of us living with the virus*

The FB counselling was also helpful in normalizing HIV, explaining that living with HIV is comparable to any lifelong disease.

*P04: Grandmother encouraged me, like she would give me an example of how a person who is diabetic lives a shorter life than you do as long as you take your medication*

*P07: They said we should not "think too much" because we are just like others who live without the virus*

Finding out their HIV status was devastating, and some young people wondered if they would survive or die soon. FB counselling reassured them that living with HIV was "not the end".

*P03: In my sessions with the grandmother I found help because she explained so many things like how I should not always be stressed, how I should not cry, and also that I should always be happy and not overthink about it [being HIV-positive]*

Lack of understanding about HIV and ART was stated as a reason to start FB counselling, as participants did not receive much information upon finding out their HIV status. FB was very helpful as participants felt more knowledgeable and empowered after the sessions.

*P10: I was expecting an explanation on why l have to take the medication because people at home they had not given me a good explanation, but after my discussion with the grandmother l really had a better understanding*

## Friendship Bench counselling on adherence to ART

Interviewees consistently reported improved ART adherence following FB counselling, and many described enhanced health and wellbeing. Most discussed how grandmothers insisted on punctuality and regularity with ART, and urged to never stop it.

*P04: I just learnt that if I'm going to collect [at the ART clinic] I should be punctual about it, even when I'm at home I should drink the pills on time after eating*

The benefits of good adherence to ART on the general health and well-being are also highlighted during counselling, together with major risks otherwise.

*P05: We spoke about how l should take my medication so that l get strong and healthy so that no one can know that l am sick [HIV positive]*

*P01: Grandmother helped me take my medication and also explained that it helps my immune system to fight against the virus and also that my viral load doesn't shoot up*

Ultimately, FB counsellors helped participants accept their HIV status and their new reality of living with HIV, and encouraged them to take their medication as prescribed.

*P05: During the sessions they tell me that if you "think too much", the soldiers in your body [CD4] decrease in numbers, that's when you start getting sick. They tell us that we should not "think too much" and just accept it [being HIV-positive] because that's what we are now and we are not alone in this. So that helped me a lot*

*P09: l thought of stopping them [. . .] the grandmothers told me that "that's life daughter just continue taking your medication" that's when l started listening to what they were saying, and l also started being healthy*

Some participants shared examples of improved health and wellbeing thanks to greater adherence to ART, like fewer headaches, weight gain, improved skin, and less hospital visits.

*P07: Previously l would always get sick and get treated with the wrong medication, but now l am fit because l'm taking the [right] medication [ART] that goes with my illness [HIV]*

## Discussion

This qualitative study aimed to explore how young people perceive the impact of FB on their adherence to ART. It also unpacked the experience of living with HIV including reasons for nonadherence to ART. Emotional denial negatively influenced adherence to ART, as these young people did not comprehend why they should be taking a lifelong treatment while they had not done anything to 'deserve' HIV.

The FB counselling seemed to have played an important role in the acceptance of HIV status, with the grandmothers offering an open and safe environment, free of judgement, where the young people were walked gently through their reality of living with HIV. Youth-specific services are not widely available across health facilities, and young people attending ART clinics are often discriminated against and judged when they actually require someone to support and reassure them, while providing them with clear information on HIV and its transmission. Only 46/47% of young women/men in Zimbabwe have comprehensive knowledge about HIV [28], showing a clear information gap on certain aspects of HIV diagnosis and treatment. At FB, these young people found a safe space where they were understood and free to talk, acquired improved knowledge about HIV and the benefits of ART, met and interacted with other YLHIV. Being told that this was not the end, seeing that they were not alone and receiving comforting words easing their self-stigma, were all contributing factors in accepting the reality of living with HIV.

FB counselling is demonstrably effective in reducing symptoms of depression [22, 29]. The association between decreased depression and improved adherence to ART has also been shown in a meta-analysis [18]. The distinctive characteristic of this study was to explore the perception of YLHIV on how FB problem-solving therapy influenced their own ART adherence. FB played a role in the critical stage of accepting HIV status, accepting the new reality of living with HIV, which ultimately led to improved adherence to ART. Encouraged by the grandmothers to take their ART as prescribed, the young people started to appreciate the immediate benefits of the medication and reported they became stronger, fitter, healthier, and relieved from the many symptoms experienced since childhood.

Whilst believing that disclosure would have benefits such as improved ART [30], parents/caregivers struggle with when and how to inform perinatally infected children of their HIV status [31, 32], and may avoid doing so. A cross-sectional study among pairs of caregivers-children in Kenya, found that the overall prevalence of disclosure was 21% for 8-year-olds, 42% for 11-year-olds and 62% among 14-year-olds [33]. Non-disclosure has negative consequences. A mixed-method study in Tanzania found that participants who discovered their HIV status on their own had significantly increased post-traumatic stress symptoms and increased internal stigma, compared with those who were purposefully told their HIV diagnosis, and were more likely to have incomplete adherence to ART [34]. The interviewees in this study described the pain that delayed disclosure caused them. A mixed-method study in Zambia insisted on the importance of commitment from parents/caregivers and health workers following serostatus disclosure to adolescents [35].

YLHIV were judgemental towards other PLHIV, alluding to promiscuity and other such behaviour, suggesting an internalised stigma around HIV [35].

## Limitations

Mental well-being and adherence to ART were self-reported and may be subject to recall bias and social desirability. The interviews taking place at the FB administrative officers may also lead to social desirability bias, with report of greater impact of the FB intervention. To limit such biases, interviewers' neutrality was critical, and participants were assured that interviews' responses would in no way impact services at the clinic. Also, 90% of interviewees acquired HIV perinatally so the views of YLHIV with more recent HIV infection may not be fully represented. However, several issues in the study such as emotional denial are not unique to perinatally acquired HIV.

Inductive thematic saturation may not have been reached, as new codes or themes might emerge if we further explore psychosocial factors of nonadherence to ART.

It is also important to discuss language and cultural barriers. Some subtleties in the feelings expressed may have been lost or some statement misinterpreted when translating from Shona to English. Study findings were discussed with the FB research assistants who conducted the interviews, to ensure no important information was omitted or misrepresented due to language and/or sociocultural context.

## Generalisability

The study was conducted in Harare, and this could limit the generalisability of the findings to rural areas of Zimbabwe, where education levels might be lower and mental health issues further stigmatised.

## Recommendations

FB problem-solving therapy is perceived by the young people as having had a great impact on their general well-being, notably through decreased symptoms of depression and improved adherence to ART. FB problem-solving therapy should be rolled out to further support young people for acceptance of HIV status and adherence to ART, and it is important that LHWs continue to play such an important role with the YLHIV and are regularly re-trained on adherence counselling.

This study highlighted a very important issue with how children and young people with perinatally acquired HIV are informed of their HIV status. Thanks to generalised access to ART, a generation is surviving into adolescence and adulthood with perinatally acquired HIV. It is important that policy and programmes take into account the lived experiences of young

people with perinatally acquired HIV and implement HIV serostatus disclosure protocols in early childhood. Counselling should be offered to parents of children living with HIV, to support them on how to discuss and disclose HIV status to their children, to reduce mental health issues and internal stigma of young people, encourage acceptance of HIV status and ultimately improve ART adherence.

## Supporting information

**S1 File. Interview topic guide in Shona.**
(DOCX)

**S2 File. Interview topic guide in English.**
(DOCX)

## Acknowledgments

The authors would like to recognize the contributions of Thembile Gola, Portia Chiuyu, Esther Gere and Abigael Muvuti who assisted with the data collection, transcription and translation.

## Author Contributions

**Conceptualization:** Ilhame Ouansafi.

**Data curation:** Ilhame Ouansafi.

**Formal analysis:** Ilhame Ouansafi.

**Funding acquisition:** Ilhame Ouansafi.

**Investigation:** Ilhame Ouansafi.

**Methodology:** Ilhame Ouansafi.

**Project administration:** Ilhame Ouansafi.

**Resources:** Ilhame Ouansafi, Epiphania Munetsi.

**Software:** Ilhame Ouansafi.

**Supervision:** Victoria Simms.

**Validation:** Dixon Chibanda, Epiphania Munetsi.

**Visualization:** Ilhame Ouansafi.

**Writing – original draft:** Ilhame Ouansafi.

**Writing – review & editing:** Dixon Chibanda, Epiphania Munetsi, Victoria Simms.

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
