## [Decision Letter · Decision Letter 0]

10 Dec 2020

PONE-D-20-17312

Impact of Friendship Bench problem-solving therapy on adherence to ART in young people living with HIV in Zimbabwe: a qualitative study

PLOS ONE

Dear Dr. Ouansafi,

Thank you for submitting your manuscript to PLOS ONE. After careful consideration, we feel that it has merit but does not fully meet PLOS ONE’s publication criteria as it currently stands. Therefore, we invite you to submit a revised version of the manuscript that addresses the points raised during the review process.

We look forward to receiving your revised manuscript.

Kind regards,

Paolo Roma

Academic Editor

PLOS ONE

Journal Requirements:

2. Please include additional information regarding the interview guide used in the study and ensure that you have provided sufficient details that others could replicate the analyses. For instance, if you developed a guide as part of this study and it is not under a copyright more restrictive than CC-BY, please include a copy, in both the original language and English, as Supporting Information.

Additional Editor Comments (if provided):

Dear Authors,

the ms certainly has merit. The suggestions provided by reviewers, especially those meant to clarify some sections (in particular regarding the sample), will strengthen the ms.

Reviewers' comments:

Reviewer's Responses to Questions

**Comments to the Author**

1. Is the manuscript technically sound, and do the data support the conclusions?

Reviewer #1: Yes

Reviewer #2: Partly

2. Has the statistical analysis been performed appropriately and rigorously? 

Reviewer #1: N/A

Reviewer #2: N/A

3. Have the authors made all data underlying the findings in their manuscript fully available?

Reviewer #1: Yes

Reviewer #2: No

4. Is the manuscript presented in an intelligible fashion and written in standard English?

Reviewer #1: Yes

Reviewer #2: Yes

5. Review Comments to the Author

Reviewer #1: The manuscript describe the impact of the FB with the use of 'grandmother' support in problem solving for adolescent who had SSQ 9 or more in baseline assessment to see how it has supported their adherence to ARVs. Adherence to ARV in adolescent and young adult is a challenge while those with mental health difficulties experience more challenges.

The manuscript is well written with a clear methodology and findings are well discussed. I do have minor observation

Title page: I think the author can put the affiliation in numbers and indicate for each authors which will be easy to follow

Outcome measure was adherence to ARVs which was judged only from the personal report which is fine as the study aimed looking at perception in relation to adherence support. However, it would have been interested to know this particular participant their SSQ-14 measures during the period of interview compared to the baseline if does correlate with the qualitative.

The intervention was no designed for HIV, it would have been good to see those with reported improved adherence and their VL if reflects the situation. This is not a scope of the paper but would add more value if the information for the few participants can be added at baseline and during interviews

The authors may have explained this in their past publication but its interested to know what age range is the so call 'grandmother' conducting the FB.

Reviewer #2: Dear Editor,

Thank you for the opportunity to review the manuscript titled “Impact of Friendship Bench problem-solving therapy on adherence to ART in young people living with HIV in Zimbabwe: a qualitative study”. This paper addresses an area of public health significance as there are few evidence-based, low-cost and potentially scalable interventions to address psychosocial needs of the large populations of young people living with HIV in sub-Saharan Africa. The paper reads well and is well supported with recent literature. However, my enthusiasm for this paper is tempered by the lack of clarity in the methods section – particularly on sample selection, which I think is inadequate. Also the paper does little to advance our understanding of the attributes of the intervention and or participants that contributed to its positive outcomes – this is important to scaling such a promising intervention.

Minor issue- there a couple of instances where citations are lacking (e.g., page 4, line 74; Page 5, line 83).

Methods

I think this section needs more information. 1) How were participants identified? 2) how would the interview setting affect the participants – this is a study assessing participants’ experiences with in intervention but interviews were conducted within their administrative offices. 3) how was the sample size determined, and was this adequate to reach saturation? More description of the data analysis with regard to quality assurance, bias reduction is also needed. The authors indicated that sampling was purposive – a description of the attributes used in this purposive sampling scheme is necessary to understand the composition of the final sample.

I think the section on study design should not focus on the parent intervention but the qualitative study design highlighting the study inclusion and exclusion criteria. If participants were selected and contacted by LHWs (who were involved in delivering the intervention??), how representative is this study population – particularly in light of the small sample size? What was the relevance of including depression as an inclusion criterion?

Findings

The papers provides an expansive description of the challenges of young people living with HIV, which is necessary to understanding the links between HIV, depression and non-adherence to ART in this population. There is also a good description of the impact of the intervention on adherence. However, I think it falls somewhat short on elucidating the aspects of this intervention that made it effective for young people – which in my opinion is critical to increasing on understanding of the relevant attributes that make such interventions successful, particularly in comparison to peer interventions.

Additionally, it would be important to understand how participant’s demographics could influence the impact of the intervention but I am not sure that the small sample size would allow for such sub-analysis.

6. PLOS authors have the option to publish the peer review history of their article (what does this mean?). If published, this will include your full peer review and any attached files.

Reviewer #1: No

Reviewer #2: No

---

## [Author Response · Author response to Decision Letter 0]

18 Feb 2021

PLOS One January 23, 2021

Dear Editor,

I would like to thank you for your revision and consideration for publication as an original research article in PLOS One of my manuscript entitled “Impact of Friendship Bench problem-solving therapy on adherence to ART in young people living with HIV in Zimbabwe: a qualitative study” by Dixon Chibanda, Epiphania Munetsi, Victoria Simms and Ilhame Ouansafi.

I read your comments and recommendations carefully, and I am hereby providing some responses and clarifications to each of the points raised.

I hope my responses below as well as the revisions in the manuscript helped clarify any remaining doubts, and I look forward to hearing from you in due course.

Sincerely, 

Ilhame Ouansafi

Tel: +212697628002

Email: ilhame.ouansafi1@alumni.lshtm.ac.uk

Part 1: Journal Requirements

Response: 

Done

 2. Please include additional information regarding the interview guide used in the study and ensure that you have provided sufficient details that others could replicate the analyses. For instance, if you developed a guide as part of this study and it is not under a copyright more restrictive than CC-BY, please include a copy, in both the original language and English, as Supporting Information.

Response:

The interview topic guide used for the semi-structured interviews is included in both Shona and English, as Supporting Information 

b) If there are no restrictions, please upload the minimal anonymized data set necessary to replicate your study findings as either Supporting Information files or to a stable, public repository and provide us with the relevant URLs, DOIs, or accession numbers. Please see http://www.bmj.com/content/340/bmj.c181.long for guidelines on how to de-identify and prepare clinical data for publication. For a list of acceptable repositories, please see http://journals.plos.org/plosone/s/data-availability#loc-recommended-repositories

Response:

The interview transcripts contain sensitive patient information related to their mental wellbeing including harm and suicidal thoughts, personal and family issues, including physical and emotional violence. It is not possible to de-identify the participants due to the small sample size and detailed history collected. The Medical Research Council of Zimbabwe (ref MRCZ/A/2130) and the London School of Hygiene and Tropical Medicine Research Ethics Committee (ref 16708) granted ethical approval for the study based on data protection. The interview files are available upon request. Transcripts have been deposited to https://datacompass.lshtm.ac.uk/ and data requests may be sent there. 

Part 2: Additional Editor Comments

Reviewer #1: The manuscript describe the impact of the FB with the use of 'grandmother' support in problem solving for adolescent who had SSQ 9 or more in baseline assessment to see how it has supported their adherence to ARVs. Adherence to ARV in adolescent and young adult is a challenge while those with mental health difficulties experience more challenges.

The manuscript is well written with a clear methodology and findings are well discussed. I do have minor observation.

Title page: I think the author can put the affiliation in numbers and indicate for each authors which will be easy to follow

Response: 

This has been addressed, to follow the standard for the journal.

Outcome measure was adherence to ARVs which was judged only from the personal report which is fine as the study aimed looking at perception in relation to adherence support. However, it would have been interested to know this particular participant their SSQ-14 measures during the period of interview compared to the baseline if does correlate with the qualitative.

Response:

As this is a qualitative study, it does not have an outcome measure as such, and it does not explore correlations.

The intervention was not designed for HIV, it would have been good to see those with reported improved adherence and their VL if reflects the situation. This is not a scope of the paper but would add more value if the information for the few participants can be added at baseline and during interviews

Response:

This is outside the scope of this paper and was not included in the ethical clearance process.

The authors may have explained this in their past publication but its interested to know what age range is the so call 'grandmother' conducting the FB.

Response:

The mean age of the grandmothers is about 53 years, as stated in the 2016 randomized controlled trial to evaluate the effect of this primary care–based psychological intervention on symptoms of common mental disorders in Zimbabwe.

(https://jamanetwork.com/journals/jama/fullarticle/2594719).

Reviewer #2: 

Dear Editor, Thank you for the opportunity to review the manuscript titled “Impact of Friendship Bench problem-solving therapy on adherence to ART in young people living with HIV in Zimbabwe: a qualitative study”. This paper addresses an area of public health significance as there are few evidence-based, low-cost and potentially scalable interventions to address psychosocial needs of the large populations of young people living with HIV in sub-Saharan Africa. The paper reads well and is well supported with recent literature. However, my enthusiasm for this paper is tempered by the lack of clarity in the methods section – particularly on sample selection, which I think is inadequate. Also the paper does little to advance our understanding of the attributes of the intervention and or participants that contributed to its positive outcomes – this is important to scaling such a promising intervention.

Minor issue- there a couple of instances where citations are lacking (e.g., page 4, line 74; Page 5, line 83).

Response:

The missing citation was added on page 4 line 74 (now page 5 line 109). On page 5 line 83 (now page 5 line 118) the reference was already there (reference 21). 

Methods

I think this section needs more information. 1) How were participants identified? 

Response:

To answer the research question, purposive sampling methodology was used to focus on particular characteristics of the population of interest, i.e. young people aged 18-24 years, living with HIV and having been prescribed ART, and having completed problem-solving therapy with FB. Eligible participants were recruited from three clinics offering FB counselling services and receiving a large proportion of adolescent patients, namely Budiriro, Glen Norah and Western Triangle.

I added a line in the manuscript in track changes, to clarify this statement. 

2) how would the interview setting affect the participants – this is a study assessing participants’ experiences with in intervention but interviews were conducted within their administrative offices. 

Response:

The interviews taking place at the FB administrative offices could indeed have influenced how participants addressed some of the questions related to impact of the FB counselling, leading to social desirability bias where interviewees might have reported greater impact of the FB counselling (for example decreased symptoms of depression or improved adherence to ART). To limit this bias, interviewers’ neutrality was critical, and the lead researcher invested significant amount of time in preparation and rehearsal with the interviewers.

I added a line in the manuscript in track changes, to clarify this statement. 

3) how was the sample size determined, and was this adequate to reach saturation?

Response:

In view of the time and resource constraints for the data collection and analysis, a sample size of 10 participants was chosen to provide a range of perspectives. Inductive thematic saturation may not have been reached, as new codes or themes might emerge if we further explore psychosocial factors of nonadherence to ART.

I added a line in the manuscript in track changes, to clarify this statement. 

 More description of the data analysis with regard to quality assurance, bias reduction is also needed. 

Response:

In exploring the perception of young people on the impact that FB counselling had on their adherence to ART, inductive analysis was used to avoid being influenced by preconceptions based on existing literature. Indeed, the predominant interest was the generation of new themes and theory emerging from the interviews.

The authors indicated that sampling was purposive – a description of the attributes used in this purposive sampling scheme is necessary to understand the composition of the final sample.

Response:

The attributes used are mentioned in the first sentence of that section, i.e. “To address the research question, semi-structured interviews were conducted with young people, aged 18-24 years, living with HIV and having been prescribed ART, who had completed the FB counselling in the last year, after initially scoring 9 or above in the SSQ-14”.

I added a line in the manuscript in track changes, to clarify this statement. 

I think the section on study design should not focus on the parent intervention but the qualitative study design highlighting the study inclusion and exclusion criteria. If participants were selected and contacted by LHWs (who were involved in delivering the intervention??), how representative is this study population – particularly in light of the small sample size? 

Response:

There wasn’t an intervention specific to young people at the time we conducted this qualitative study, and participants were selected among young people having completed the FB counselling in the last year. Even if participants were contacted by FB LHWs, the study population is still representative as the study is targeting young people who have completed the FB intervention, specifically. 

What was the relevance of including depression as an inclusion criterion?

Response:

FB problem-solving therapy is intended to people suffering from common mental illnesses, including depression. There is significant literature on the association of depression and adherence to ART, and this study aimed to complement such literature by understanding qualitatively some of the psychosocial factors contributing to nonadherence to ART among young people living with HIV.

Findings

The papers provides an expansive description of the challenges of young people living with HIV, which is necessary to understanding the links between HIV, depression and non-adherence to ART in this population. There is also a good description of the impact of the intervention on adherence. However, I think it falls somewhat short on elucidating the aspects of this intervention that made it effective for young people – which in my opinion is critical to increasing on understanding of the relevant attributes that make such interventions successful, particularly in comparison to peer interventions.

Response:

The FB counselling seemed to have played an important role in the acceptance of HIV status, the young people needed a friendly soul to walk them gently through what they already knew, a supportive person who would provide them with information on HIV and its transmission in a way that is understandable to them, someone to reassure them. Young people were comfortable discussing with the grandmothers as they felt safe, understood and not judged. With FB, these young people found a safe space where they are free to talk, they acquired improved knowledge about HIV and the benefits of ART, they met and interacted with other people living in similar or sometimes worse conditions, and they were told that they were not guilty for having been infected with HIV. 

I added a line in the manuscript in track changes, to clarify this statement. 

Additionally, it would be important to understand how participant’s demographics could influence the impact of the intervention but I am not sure that the small sample size would allow for such sub-analysis.

Response:

The small sample size does not allow it. Furthermore, while it could have been interesting to have a short demographic survey to capture standardized information about the participants (such as community, education level, employment or marital status), not having that information removed potential for deductive disclosure.

---

## [Decision Letter · Decision Letter 1]

31 Mar 2021

Impact of Friendship Bench problem-solving therapy on adherence to ART in young people living with HIV in Zimbabwe: a qualitative study

PONE-D-20-17312R1

Dear Dr. Ouansafi,

We’re pleased to inform you that your manuscript has been judged scientifically suitable for publication and will be formally accepted for publication once it meets all outstanding technical requirements.

Kind regards,

Paolo Roma

Academic Editor

PLOS ONE

Additional Editor Comments (optional):

Reviewers' comments:

Reviewer's Responses to Questions

**Comments to the Author**

1. If the authors have adequately addressed your comments raised in a previous round of review and you feel that this manuscript is now acceptable for publication, you may indicate that here to bypass the “Comments to the Author” section, enter your conflict of interest statement in the “Confidential to Editor” section, and submit your "Accept" recommendation.

Reviewer #1: All comments have been addressed

Reviewer #2: All comments have been addressed

2. Is the manuscript technically sound, and do the data support the conclusions?

Reviewer #1: Yes

Reviewer #2: Yes

3. Has the statistical analysis been performed appropriately and rigorously? 

Reviewer #1: N/A

Reviewer #2: Yes

4. Have the authors made all data underlying the findings in their manuscript fully available?

Reviewer #1: Yes

Reviewer #2: No

5. Is the manuscript presented in an intelligible fashion and written in standard English?

Reviewer #1: Yes

Reviewer #2: Yes

6. Review Comments to the Author

Reviewer #1: The authors have responded to the comments I had raised before, the manuscript is suitable for publication, I recommend it to be accepted

Reviewer #2: The authors have addressed my previous concerns regarding the manuscript. I don't have any additional comments.

7. PLOS authors have the option to publish the peer review history of their article (what does this mean?). If published, this will include your full peer review and any attached files.

Reviewer #1: **Yes: **Blandina T Mmbaga

Reviewer #2: No

---

## [Editor Report · Acceptance letter]

12 Apr 2021

PONE-D-20-17312R1 

Impact of Friendship Bench problem-solving therapy on adherence to ART in young people living with HIV in Zimbabwe: a qualitative study 

Dear Dr. Ouansafi:

I'm pleased to inform you that your manuscript has been deemed suitable for publication in PLOS ONE. Congratulations! Your manuscript is now with our production department. 

Kind regards, 

on behalf of

Prof. Paolo Roma 

Academic Editor

PLOS ONE